# Nanotechnology-Based Drug Delivery Approaches of Mangiferin: Promises, Reality and Challenges in Cancer Chemotherapy

**DOI:** 10.3390/cancers15164194

**Published:** 2023-08-21

**Authors:** Muhammad Sarfraz, Abida Khan, Gaber El-Saber Batiha, Muhammad Furqan Akhtar, Ammara Saleem, Basiru Olaitan Ajiboye, Mehnaz Kamal, Abuzer Ali, Nawaf M. Alotaibi, Shams Aaghaz, Muhammad Irfan Siddique, Mohd Imran

**Affiliations:** 1College of Pharmacy, Al Ain University, Al Ain P.O. Box 64141, United Arab Emirates; 2Department of Pharmaceutical Chemistry, Faculty of Pharmacy, Northern Border University, Rafha 91911, Saudi Arabia; aqua_abkhan@yahoo.com; 3Department of Pharmacology and Therapeutics, Faculty of Veterinary Medicine, Damanhour University, Damanhour 22511, AlBeheira, Egypt; 4Riphah Institute of Pharmaceutical Sciences, Riphah International University Lahore, Lahore 54000, Pakistan; 5Department of Pharmacology, Faculty of Pharmaceutical Sciences, GC University Faisalabad, Faisalabad 38000, Pakistan; 6Phytomedicine and Molecular Toxicology Research Laboratory, Department of Biochemistry, Federal University Oye-Ekiti, Oye 371104, Ekiti State, Nigeria; bash1428@yahoo.co.uk; 7Department of Pharmaceutical Chemistry, College of Pharmacy, Prince Sattam Bin Abdulaziz University, Al-Kharj 11942, Saudi Arabia; 8Department of Pharmacognosy, College of Pharmacy, Taif University, Taif 21944, Saudi Arabia; 9Department of Clinical Pharmacy, College of Pharmacy, Northern Border University, Rafha 91911, Saudi Arabia; 10Department of Pharmacy, School of Medical & Allied Sciences, Galgotias University, Greater Noida 203201, India; 11Department of Pharmaceutics, Faculty of Pharmacy, Northern Border University, Rafha 91911, Saudi Arabia

**Keywords:** mangiferin, xanthone, nanoparticles, bioavailability, cancer, phytomedicine

## Abstract

**Simple Summary:**

The current review describes mangiferin (MGF), a secondary plant metabolite of a xanthone derivative obtained from mango plants and known for its therapeutic activities. It is potent against various cancers, including breast, liver, prostate and glioblastoma, by inhibiting lipid peroxidation and preventing NFκB activation. However, MGF faces challenges such as poor lipophilicity, high first-pass metabolism and extensive P-gp efflux. Nano-drug delivery and deep learning-based approaches are being explored to overcome these challenges. Clinical trials and patents are ongoing, and MGF is expected to become a promising therapeutic biological molecule for cancer treatment.

**Abstract:**

Mangiferin (MGF), a xanthone derived from *Mangifera indica* L., initially employed as a nutraceutical, is now being explored extensively for its anticancer potential. Scientists across the globe have explored this bioactive for managing a variety of cancers using validated in vitro and in vivo models. The in vitro anticancer potential of this biomolecule on well-established breast cancer cell lines such as MDA-MB-23, BEAS-2B cells and MCF-7 is closer to many approved synthetic anticancer agents. However, the solubility and bioavailability of this xanthone are the main challenges, and its oral bioavailability is reported to be less than 2%, and its aqueous solubility is also 0.111 mg/mL. Nano-drug delivery systems have attempted to deliver the drugs at the desired site at a desired rate in desired amounts. Many researchers have explored various nanotechnology-based approaches to provide effective and safe delivery of mangiferin for cancer therapy. Nanoparticles were used as carriers to encapsulate mangiferin, protecting it from degradation and facilitating its delivery to cancer cells. They have attempted to enhance the bioavailability, safety and efficacy of this very bioactive using drug delivery approaches. The present review focuses on the origin and structure elucidation of mangiferin and its derivatives and the benefits of this bioactive. The review also offers insight into the delivery-related challenges of mangiferin and its applications in nanosized forms against cancer. The use of a relatively new deep-learning approach to solve the pharmacokinetic issues of this bioactive has also been discussed. The review also critically analyzes the future hope for mangiferin as a therapeutic agent for cancer management.

## 1. Introduction

Xanthones are the dibenzo-pyrone tricyclic secondary metabolites having a symmetric structure [1]. The Greek word for the compounds’ yellow color, “xanthos”, refers to these yellow compounds. Their structure is numbered according to the development of their biosynthetic processes [2,3]. As a result, the first two rings produced from the acetate and shikimate pathways, respectively, are assigned to carbons 1–4 and 5–8. The numbering is in accordance with the International Union of Pure and Applied Chemistry (IUPAC’s) guidelines for the chemical 9H-xanthen-9-one (Figure 1) [1,4]. Xanthones are called “promiscuous binders” due to their capacity to bind to a wide range of receptors and exhibit great pharmacological activity against several illnesses [5,6]. Because of this, the research of synthetic xanthone derivatives has been prioritized by organic chemists to develop novel therapeutic candidates that have, among other things, anti-inflammatory, antioxidant and anticancer properties [7].

Previous studies have discussed xanthones numerous times due to their significant role in natural products. Recently, xanthones have been emphasized as a possible source of anticancer substances with high oral bioavailability [8]. There are only four families and six genera of plants, *Swertia* L. and *Gentiana* L. from Gentianaceae, *Platonia MART. JUSS.* and *Garcinia* L. from Clusiaceae, *Calophyllum* L. from Calophyllaceae and J. AGARDH (=Gutifferae JUSS.), *Mangifera* L. from the Anacardiaceae family that were initially found to contain xanthones [9,10]. This class was anticipated to be helpful as chemo-systematic indicators due to its restricted distribution. Recent research has shown that xanthones are, in fact, extensively distributed in large plants, which restricts the utility of xanthones as chemo-systematic indicators [11].

Mangiferin is a C-glucosyl xanthone that mangoes contain in large quantities. Due to its extensive cultivation range, demand and taste, mango is often referred to as the king of all fruits. Mango is also an essential component of traditional medicine for numerous indigenous communities. According to Vyas et al. (2012), mangoes and cashews are members of the Anacardiaceae family, which contain poisonous plants like poison ivy and oak [12,13]. The fruit grows on the *Mangifera indica* L. tree, native to the Asian subcontinents and other temperate regions, and has been cultivated in India for more than 4000 years [14]. After being established in East Africa, it is thought to have traveled from East Asia to China, the West Indies, the Caribbean, Brazil, Mexico and the United States between the fourth and fifth centuries. The term “mango” is derived from the Tamil word “mangkay” or “man-gay,” which Portuguese later took and changed to its English equivalent. Aphrodisiac or regular uses of mangoes such as treating stomach disorders are mentioned in several Indian texts in addition to their nutritional value [15].

MGF, a mango derivative, is a glucosyl xanthone and polyphenolic antioxidant. It has demonstrated antioxidant properties and can be utilized to stop the peroxidation of some lipids. Additionally, it functions as an antidiabetic, cardiotonic, anti-degenerative and wound healer [16]. MGF is a powerful component found in over 40 polyherbal formulae used in traditional Chinese medicine [17]. Since then, it has been detected in numerous species across 16 distinct families, including Gentianaceae, Anacardiacae and Iridaceae [18]. It was initially identified in the *M. indica*. The plant is used extensively in religious and social ceremonies and for nutrition and therapeutic purposes. According to Stohs et al. (2018), the leaves are employed in Hinduism on practically all significant occasions, including worship and wedding rituals [19]. According to Laurindo et al. (2015), the fruit’s outer peels are used as cosmetics, and Namngam and Pinsirodom (2017) reported that the seeds are used to make beauty creams. The control of biogenic amines has been proposed as a potential safety profile for MGF, which has recently been discovered to have antidepressant characteristics (Table 1) [20,21]. MGF has been demonstrated to pass the blood–brain barrier and lower oxidative stress in neurodegenerative diseases. All four of Lipinski’s criteria, i.e., the molecular mass of less than 500 Da, high lipophilicity, less than five hydrogen bond donors and fewer than ten hydrogen bond acceptors, are satisfied by the chemical makeup of MGF when taken orally. Numerous pathophysiological traits of MGF imply that it might make an effective therapeutic [22].

MGF is a potent antioxidant with incredible therapeutic benefits, including analgesic, antiviral, anticancer, antidiabetic, antioxidant, antiaging and immunomodulatory activities [23]. The therapeutic profile of MGF has been described briefly in Figure 2 [24]. The phenolics homomangiferin and isomangiferin, which comprise a subsequent percentage of all phenolics, are also found in the mango tree’s twigs, leaves and mango peel. They can chelate iron in Fenton-type reactions, preventing hydroxyl radicals’ formation [25]. Methanolic MGF extracts have been shown to protect the liver against carbon tetrachloride-induced liver damage, further demonstrating the effectiveness of their in vivo radical scavenging system [26,27]. MGF significantly reduced lipid peroxidation brought on by hydrogen peroxide in a dose-dependent manner in human peripheral blood cells. According to a study, MGF (1, 10 and 100 g/mL) treatment significantly enhanced the erythrocytes’ resistance to hydrogen peroxide-induced reactive oxygen species (ROS) [28].

Additionally, its dose-dependence reinstates energy charge potential in hydrogen peroxide-treated red blood cells and prohibits the loss of guanosine triphosphate (GTP), adenosine triphosphate (ATP) and total nucleotide (NT) damage. After initiating the MGF iron complexes and scavenging free radicals, it protects hepatocytes from the destruction caused by free radical-mediated hypoxia/reoxygenation injury [29,30]. The rat livers treated with 50 M iron citrate and 10 M MGF prevented the loss of enlargement and trans-membrane potential of mitochondria. The more stable iron complex forms preferentially in the absorption spectra of MGF. The substance cannot participate in the above-mentioned mechanisms for lipid peroxidation and Fenton-type reactions [31].

This review looked at and summarized the history, sources and structure elucidation of MGF available in the previous literature to support the future potential of MGF as a novel therapeutic drug for cancer. The review mainly focused on the mechanism of MGF in different types of cancers and the recent approaches in MGF-based nanocarriers that have successfully increased MGF solubility and bioavailability in cancer cells.

## 2. Mangiferin

### 2.1. History and Discovery of Mangiferin

The first pure MGF was produced by *Mangifera indica* L. in 1908. MGF’s molecular structure was determined to be 1,3,6,7-tetrahydroxy-xanthone 2-C-D-glucopyranoside by Roberts and Nott in 1967 [32]. Two years later, the structure of isomangiferin, 1,3,6,7-tetrahydroxyxanthone 4-C-D-glucopyranoside (1970) was reported, and Aritomi and Kawasaki (1969) discovered homomangiferin. In 1985, a report was published based on MGF’s studies on X-ray crystallography and nuclear magnetic resonance spectra [33]. In particular, the Chiretta plant, Trichomanes reniforme G, finished the biochemical synthesis of MGF, isomangiferin and homomangiferin in 2010 [34]. Nevertheless, before that, Nott and Roberts created MGF in the form of glycone from 1,3,6,7-tetrahydroxyxanthone while it was present in acetobromoglucose [32].

### 2.2. Source of Mangiferin

MGF has piqued researchers’ interest because of its wide range of pharmacological properties and ease of accessibility. It has encouraged them to create effective extraction and isolation techniques that could provide the highest yield possible. The mango tree, which grows in abundance in nature, is the primary source of MGF [35]. The capacity of this plant with substantial nutritional and medicinal benefits, connected to diverse families such as Anacardiaceae, Podalyfrieae, Hypericoideae, etc., is discussed in numerous ancient texts. According to reports, the average amounts of MGF detected in different mango tree parts are 42 mg/kg in seed kernels, 4.4 mg/kg in mango pulp, 71.4 g/kg in stem bark and 1690 mg/kg in mango peel [36]. Because of this, using innovative methodologies has a great chance of producing the highest output for larger-scale exploration at the lab and industrial levels [37].

### 2.3. Extraction, Isolation and Structural Elucidation of Mangiferin

One of the primary sources of MGF is *Mangifera indica*. It is found in the solvent extracts of the bark, peels, kernels and old and young leaves. There are several reports of the extraction of MGF from the different parts of the *Mangifera indica*. Baretto et al. discovered and measured the secondary metabolites of this plant. The procedure is to extract the *Mangifera indica* plant material with solvent, i.e., hexane, in a Soxhlet system to elucidate lipids. After drying, the substance was extracted with methanol, and the resulting solutions were vaporized using a rotatory evaporator [38]. Various analysis techniques were used to analyze the extracts after they had been dissolved in methanol. A hypoxanthine/xanthine oxidase assay based on high-performance liquid chromatography (HPLC) was used to assess the solvent extracts’ capacity to scavenge free radicals, and the results showed that all of the extracts had dose-dependent antioxidant capacity. Additionally, other in vitro assays, such as oxygen radical absorbance capacity and the plasma’s capacity to reduce ferric ions, were used to assess the significant phenolic compounds such as methyl gallate, MGF and penta-O-galloylglucoside gallic acid found in these extracts [39].

Nong et al. analyzed the MGF obtained from *Mangifera persiciformis* leaves and *Mangifera indica’s* bark using capillary zone electrophoresis (CZE). The chemical showed extraordinarily significant antioxidant capabilities [40]. MGF was isolated from *Mangifera persiciformis* leaves using a vacuum evaporation process after the leaves were extracted with absolute ethanol for a day at room temperature. After achieving the visibility of golden-colored MGF (m.p. 271–273 °C), the condensed residue was resuspended in distilled water and subjected to rotatory vacuum evaporation with petroleum ether and ethyl acetate successively. The chemical produced in this way has a purity of about 97.39%. MGF was separated from an ethanol and water combination [40]. DaCruz et al. published the single-crystal X-ray structure of the MGF [33]. Nunez Selles et al. identified the main active ingredient in MGF from the water-based decoction of the stem bark of *Mangifera indica.* They qualitatively examined it using HPLC [41]. From the peels of *Mangifera indica*, Schieber and colleagues extracted 14 flavonol and xanthone glycosides and then used HPLC-electrospray ionization mass spectrometry to analyze the substances [42]. MGF, isomangiferin and their derivatives were discovered to be the xanthone glycosides based on the fragmentation pattern of the mass spectra. Gowda et al. developed and validated a reversed-phase liquid chromatographic technique for detecting MGF in alcoholic extracts of *Mangifera indica* at a wavelength of 254 nm and found the presence of MGF in the extract [12].

### 2.4. Structure Elucidation of Mangiferin

MGF is difficult and expensive to synthesize chemically. Hence, the best way to obtain it is to extract it from various biological sources. MGF is a C2-d-glucopyranosyl-1,3,6,7-tetrahydroxyxanthone chemical. The aromatic ring that is connected to the C-C bond of a glucose moiety in C-glucosyl xanthone’s chemical structure imparts its high-water solubility and polarity [31]. The redox-active aromatic systems, catechol rings and free hydroxyl groups brought on by the xanthone moiety give MGF its antioxidant properties. It also has effective iron-chelating properties, preventing hydroxyl radicals from entering the Fenton-type reactions primarily responsible for oxidation [26]. Along with MGF, the mango tree accommodates the other isomeric forms, such as isomangiferin and homomangiferin. The chemical name of isomangiferin, which is mostly found in *Anemarrhena asphodeloides*, is 4-d-glucopyranosyl-1, 3, 6, 7-tetrahydroxy- 9H-xanthen-9-one, and homomangiferin is 1, 6, 7-trihydroxy-3-methoxy 2-c-d-glucopyranosyl-xanthone (Figure 3) [37].

## 3. Molecular Mechanism of Action of Mangiferin in Cancer

MGF acts through numerous mechanisms to exhibit anti-inflammatory, anti-apoptotic, immunomodulatory, anti-genotoxic, cell cycle arrest and antiproliferative effects, collectively exerting anti-tumor activity [43,44]. The in vitro and in vivo studies on various malignancies have shown that MGF has a broad spectrum of effectiveness [45,46]. According to available data, MGF’s adverse effects range from minor to non-existent; nevertheless, there might be some variance depending on the MGF’s source [47]. Figure 4 depicts mangiferin’s molecular targets and biomarkers [48].

### 3.1. Inflammation

It is widely believed that the continuous stimulation of inflammatory processes contributes to the development of cancer [49]. Chronic inflammation, which could be brought on by viral or bacterial infections, autoimmune disorders or ongoing irritation, is thought to be responsible for 20% of malignancies [50]. Chronic inflammation can fuel the growth of tumors by creating a permissive surrounding. Additionally, inflammation produces ROS, which can damage DNA and increase a substance’s ability to cause cancer. MGF is hypothesized to primarily interfere with activated B cells’ nuclear factor light chain enhancer (NFκB) to suppress the inflammatory response [51]. In addition to creating unfavorable conditions for cancer, MGF has been shown to have antidiabetic properties and lower the danger of heart disorders. MGF also lowers cholesterol and glucose levels in serum, lessening the severity and progression of heart diseases and diabetes [52]. Hence, MGF anticipates several benefits for cardiovascular disease, cancers and diabetes. In contrast, various drugs help treat all these ailments, which might form adverse effects on the human body and lead to other disorders. Some pathways via which MGF acts on cancer cells have been described here [53].

### 3.2. Nuclear Factor k-Light-Chain-Enhancer of Activated B Cells Activity

The expression of growth factors, migratory molecules, pro-inflammatory cytokines and other genes that take part in survival and proliferation are just a few of the crucial processes that NFκB controls in inflammation [54]. NFκB is activated when there is inflammation. The pictorial representation of the inhibition of NFκB by MGF has been shown in Figure 5 [55]. The interleukin-1 receptor-activated kinase 1 (IRAK1) has been impressed onto this receptor-signaling complex by myeloid differentiation primary response gene 88 (Myd88) for phosphorylation after the ligand binding and activation of Interleukin-1 receptor (IL-1Rs), and Toll-like receptors (TLRs) in inflammatory conditions. [54,55]. The subsequent autophosphorylation and phosphorylation by IRAK4 are made possible by IRAK1′s association with Myd88. IRAK1 forms a compound with tumor necrosis factor receptor-associated factor 6 (TRAF6) in its phosphorylated state that sends signals through NFκB and is eventually activated by NEMO/IKK- β /IKK-α, subunit- β /Inhibitor of NFκB Kinase subunit-α, NFκB essential modulator/suppression of NFκB Kinase and suppression of kB (IkB)/p65/p50 complexes [56,57]. According to recent research, MGF blocks NFκB activation at different points along the route. Both traditional and non-classical pathways can activate NFκB. The p50 and IkB kinase complex control the conventional tracks, whereas IKKα and p52 control the alternative pathways [58].

### 3.3. Initial Stimulus for NFκB Activation

After testing by several researchers, it was discovered that MGF inhibits ROS formation to block NFκB activation caused by lipopolysaccharide (LPS), peptidoglycan (PDG), hydrogen peroxide (H_2_O_2_) phorbol 12-myristate-13-acetate (PMA) or tumor necrosis factor (TNF) [58]. The IRB3 AN27 (human fetal neural) cell lines, U-937 (lymphoma), MCF-7 (breast cancer) and HeLa (cervical cancer) have all shown this effect [45]. In peritoneal macrophages stimulated by LPS and PDG, Jeong et al. showed that the suppression impact of MGF on the expression of NFκB was partially caused by suppression of the phosphorylation and activation of IRAK1. MGF prevents NFκB activation concurrently by inhibiting inflammatory genes [59,60]. MGF’s inhibition of IRAK1 may prevent the emergence of chemotherapeutic drug resistance. In particular, IRAK1 overexpression has been linked to triple-negative breast tumors, and it has been suggested that the paclitaxel resistance might be reversed by inhibiting the IRAK1 via the p38-MCL1 pathway [61]. MGF is used in combination therapies. Several studies suggested the suppression of TNF signal transduction by MGF, the cationic interaction between the TNFR-associated factor 2 (TRAF2), tumor necrosis factor receptor type-1-associated death domain protein (TRADD) and NCK Interacting Kinase (NIK), with the TNF Receptor (TNFR) along with the initiation of NFκB activation via deterioration and phosphorylation of IkB [62,63]. The cell lines named U-937 were co-expressing with plasmids for TRAF2, TNFR1, IKK, p65, TRADD and NIK to determine the location of the action [64]. The secreted embryonic alkaline phosphatase (SEAP) was employed as a gene for NFκB, and the expression magnitude in untreated and treated cells was tracked. MGF reduced the expression of SEAP induced by TRAF2, TNFR1, IKK, p65, TRADD and NIK, but it had no appreciable impact on SEAP induced by p65. MGF must therefore function after IKK [55,65].

### 3.4. Angiogenesis

Sustained angiogenesis is considered a distinctive characteristic of cancer [66]. Hence, angiogenic tumors can grow and spread using nutrients and oxygen from their blood supply. It is well known that the VEGF-A protein promotes angiogenesis. Vimang^®^ extracts and MGF have shown repressive effects on TNF-induced VEGF-A transcription in metastatic breast cancer MDA-MB-231 cell lines [67]. The duration of this trial was, however, relatively brief. To better understand how MGF affects angiogenesis, longer-term research and data from in vivo investigations are needed [67].

### 3.5. Proliferation/Metastasis

The intermediates of these processes may be dysregulated in cancer cells, which enables the proliferation of cancer cells to outpace cell death rates because cell adhesion pathways are dysregulated, and cancer cells may become more motile. The loss of sticking enables cells to expand beyond the original place of inception and results in secondary cancers. MGF inhibits cell division by altering β-catenin, affecting epithelial-to-mesenchymal transition (EMT), MMP-7 and MMP-9. MGF may affect VEGF-A receptor transcription through NFκB to control angiogenesis. MGF has also effectively inhibited various interlinked signal transduction pathways in breast cancer cells [67].

### 3.6. Apoptosis

Cancer cells can avoid apoptosis despite having malignant features to live and grow. Apoptosis can usually be induced by the death receptors involving the extrinsic or intrinsic pathways through mitochondria. Despite this, the extrinsic pathway occurs by procaspase-8 and Fas-associated death domain (FADD), and the intrinsic pathway typically entails increased membrane permeability of mitochondria [68]. The favored cell death method is called apoptosis because necrotic cell death might result in inflammatory alterations. After all, immune-stimulating chemicals are released. Many chemotherapeutic agents aim to cause malignant cells to undergo apoptosis to eradicate cancer. From past research, it can be inferred that MGF regulates apoptosis via a variety of sites and expresses favorable apoptosis-inducing capabilities in a variety of cells [8,14,36], causing cancerous cells to undergo apoptosis (Figure 6) [55].

### 3.7. Other Anticancer Pathways

MGF has anti-inflammatory benefits and anticancer effects by controlling several signaling pathways (such as the Notch3 and Akt pathways) linked with apoptosis, proteins and other inflammatory factors. MGF decreases the proliferation of human ovarian cancer cells by modifying the notch three pathway and the yes-associated protein (YAP) pathway [69]. After signaling to the Rac1/Wiskott–Aldrich syndrome verprolin-homologous protein-2, the migration of breast cancer cells and invasion were reduced by MGF. MGF also caused apoptosis and inhibited growth in breast cancer cells by blocking the mevalonate pathway [70]. MGF was discovered to cause cell death and decrease TPC-1 viability by downregulating Bcl-2 expression and upregulating cas-3. The PI3K/Akt pathways were suppressed by MGF, significantly reducing gastric cancer cell proliferation and promoting apoptosis [71]. MGF increased the expression of cleaved caspase-9, Bax, Bad and caspase-3 and lowered the expression of Mcl-1, Bcl-2 and Bcl-xL genes in SGC-7901 cells. According to Tan et al., MGF inhibited the growth of hepatocellular carcinoma by controlling the suppression of the WT1-associated lymphoid enhancer-binding factor 1 (LEF1) by Wnt signaling [72], additionally, by enhancing the DNA damage after radiation and reducing the proliferation. MGF may make glioblastoma multiforme cells more sensitive to radiotherapy [73]. MGF downregulated MMP2 and MMP9, significantly slowing the growth of human epithelial ovarian cancer cells. According to several studies, bacteria can influence the development of cancer and inflammation, and their interactions with host cells can prevent cancer development. The regulation of the flora by environmental factors has some bearing on cancer development. A brief about the mechanisms of MGF towards cancer cells has been mentioned in Table 2 [38]. However, little research has been done on how MGF controls cancer-related microbes [72].

## 4. Mangiferin: Challenges in the Clinical Translation

MGF has several drawbacks, such as low bioavailability and poor solubility like other phytoconstituents. The oral bioavailability of MGF is reported to be less than 2% [81,82] only, and water solubility is also 0.111 mg/mL [83]. These two issues with MGF restrict its clinical application to cancer. Despite this, MGF also shows poor lipophilicity, i.e., log P of -0.56, high first-pass metabolism and extensive P-gp efflux with metabolism by gut Cytochrome P-450 enzymes which results in the less amount of drug at the disease sites to show substantial therapeutic efficacy. The low aqueous solubility and lipophilicity make the MGF a Biopharmaceutical Classification System (BCS) class IV agent [84].

### 4.1. Need for Novel Drug Delivery Systems of Mangiferin

MGF has low bioavailability and poor water solubility, so encapsulation in nanosystems might be a valuable method to increase these properties (1.2% in rats). The peak plasma concentration occurred at about one hour. It was unexpectedly modest given the high levels of compounds that are less water soluble, such as MGF, which is most suited for encapsulation since it improves the PK characteristics in general. The retention of MGF in the nanoparticles is better in pectin-based formulations than chitosan-based formulations when prepared using the spray drying process [84,85]. There have been many different kinds of nanovehicles created, and many polysaccharide-based ones have been utilized to carry anticancer medications. Few of them might interrelate with membrane receptors [86,87]. To deliver MGF, specialized nanovehicles made of polysaccharides may be appropriate [16]. There is little doubt that additional research is needed to increase MGF’s bioavailability and transport mechanisms from other supplements to cancer sites. A “smart vehicle” will probably need to be specifically tailored to MGF and perhaps the kind of cancer to deliver the drug to cancer cells instead of healthy ones and avoid or reduce a delivery gradient in the solid tumor [54]. The nanocarriers used to deliver MGF are presented in Figure 7 [16,88]. The mixed micelles are developed using amphiphilic block copolymer. The main advantage of the mixed micelles is their ability to encapsulate the lipophilic drug in their hydrophobic core, and the outer hydrophilic shell helps the mixed micelle stability in an aqueous environment [89,90,91]. The nanocapsules are versatile mixed vesicular nano-drug delivery systems, usually with a lipid core and polymeric wall. The nanocapsules benefit the therapeutic and imaging agents for diseases such as cancer [92].

On the other hand, nanoparticles are a broad range of materials that contain particulate substances of less than 100 nm in size. The importance of these nanoparticles was realized when it was found that the size greatly impacts a material’s physiochemical properties. These comprise the surface layer, the shell and the core. The nanoparticles are divided into several categories, such as carbon, lipid-based or polymeric nanoparticles [93]. Gold nanoparticles are one of the categories of nanoparticles and have received the most attention in today’s era because of their biocompatible nature, stability and ease of preparation. Gold contains a high number of electrons and very low chemical reactivity. Due to all these benefits, gold nanoparticles are appropriate for imaging agents and manifest optical characteristics such as scattering of absorbance in the near-infrared region [94]. Carbon nanotubes have a characterized nano hollow tube-shaped structure and can be utilized for drug delivery, tumor imaging and photothermal therapy. CNTs can actively or passively target malignant tumor cells [95].

### 4.2. Nanocarriers of Mangiferin for Cancer Management

For the synergistic action of doxorubicin (DOX), Aboyewa et al. (2021) synthesized gold nanoparticles (AuNPs) using cyclopia intermedia extract. According to the study, an extract of *C. intermedia* was used to synthesize AuNPs for the first time. The research suggested a potential synergistic relationship between AuNPs (HB-AuNP) and DOX made from *C. intermedia* and between DOX and AuNPs (MGF-AuNP) made from MGF. MGF-AuNPs and HB-AuNPs have comparable physicochemical properties and biological activity. MGF may be involved in the synthesis of HB-AuNPs, given that the extract of *C. intermedia*, which has a high concentration of MGF, was used to make HB-AuNPs. The effect of the biogenic AuNPs and DOX on PC-3, Caco-2, MCF-12A and U87cells was determined. The HB-AuNPs and the MGF-AuNPs showed similar toxicity toward the four cell lines. At all the concentrations, the percent cell viability of MCF-12A cells was higher or near 100% for both the formulations, i.e., MGF-AuNPs (15.62–125 µg/mL) and HB-AuNPs (15.62–250 µg/mL). These results exhibit that the AuNPs at these concentrations can protect the cells and could enhance cell proliferation. It is also possible that MGF, linked to multiple studies suggesting that it can increase the anti-tumor effects of DOX, is responsible for the synergistic relationship between DOX and MGF-AuNPs and between DOX and HB-AuNPs. These findings can provide hope for a better cytotoxic effect of MGF after delivering through nanoparticles [96].

Wang et al. (2023) developed hyaluronic acid (HA) modified mangiferin–methotrexate-loaded nanoparticles by a self-assembly method. The findings showed that MGF functions as an anti-inflammatory agent and that methotrexate can be employed as a tumor-targeting ligand of the folate receptor (FA) and HA as another tumor-targeting ligand of the CD44 receptor. The ester bond was effective at coupling HA, MGF and methotrexate together, according to 1HNMR and FT-IR data. The size of the hyaluronic acid (HA) modified mangiferin–methotrexate-loaded nanoparticles was determined and found to be around 138 nm. Hyaluronic acid (HA)-modified mangiferin–methotrexate-loaded nanoparticles were shown through in vitro cell tests to have a practical inhibitory effect on K7 cancer cell lines with proportionally lesser toxicity to healthy MC3T3-E1 cell lines than methotrexate. These findings demonstrated the ability of the prepared hyaluronic acid (HA)-modified mangiferin–methotrexate-loaded nanoparticles to selectively ingest K7 cancer cells through endocytosis from FA and CD44 receptors, thereby blocking the development of tumor tissues and lowering the nonspecific cellular uptake toxicity brought on by the therapy from these cancer drugs [97].

Yasiri et al. (2017) prepared radioactive gold nanoparticles loaded with mangiferin (MGF-198AuNPs) for prostate tumor therapy. The MGF-198AuNPs are created when MGF interacts with an Au-198 gold precursor. Au-198′s beta emissions have particular benefits for treating tumors, and gamma rays are utilized to quantify the amount of gold present in tumors and other organs. MGF’s affinity for laminin receptors allows for the selective accumulation of therapeutic payloads of this novel chemotherapeutic drug in prostate cancer cells (PC-3) derived from human prostate tumors in mice which overexpress this receptor subtype. Through the intratumoral delivery of MGF-198AuNPs, extensive in vivo therapeutic efficacy studies demonstrated a retention of over 80% of the injected dose (ID) in prostate tumors for up to 24 h. After the treatment for three consecutive weeks, the treated group of animals had tumor volumes that were over five times lower than those of the saline-treated group. A new green technology and paradigm to prepare mangiferin-loaded nanoparticles could be a potential way to treat cancer in oncology [98].

Khoobchandani et al. (2021) synthesized mangiferin-loaded gold nanoparticles (MGF-AuNPs) for the immunomodulatory intervention in prostate cancer. Through observations, it was found that there is an increase in levels of anti-tumor cytokines, i.e., IL-12 and TNF-α, and concurrent decreases in levels of pro-tumor cytokines, i.e., IL-10 and IL-6. The study exhibited the immunomodulatory intervention of MGF-AuNPs in prostate cancers. TNF- α and IL-12 increased nearly ten times in the MGF-AuNP-treated groups, while IL-10 and IL-6 decreased by around two times. Targeting the NF-kB signaling pathway causes MGF-AuNPs to be able to target splenic macrophages. This nanotechnology-based nanomedicine could be an environmentally friendly immunomodulatory intervention for the therapeutic effectiveness of MGF in the prostate [99].

Razura-Carmona et al. (2023) prepared poly-lactic-co-glycolic acid-functionalized nanoparticles loaded with MGF and lupeol to enhance the therapeutic efficacy against prostate cancer. The best treatment demonstrated behaviors as a topoisomerase II inhibitor, according to the cell-based tests of nanoparticles that showed controlled release and encapsulation efficiency of lupeol (60.01% and 1.24%) and MGF (57.7% and 1.94%). The nanoparticles created in this study had no harmful effects on BEAS-2B cells. However, they did reduce the viability of HepG2 cells with IC50 of 1549.9% at 174.62 g/mL of concentration. On the contrary, morphological changes brought on by TF15 are noticed at doses of 2500 and 1250 g/mL, even though the hemolytic activity is not visible after one hour of treatment. The TF15 treatment demonstrated no cytotoxicity effect for healthy cells, maintained biological activity and slowed the proliferation of cancer cells [100].

Khurana et al. (2018) developed vitamin-E TPGS-based mangiferin-loaded phospholipid mixed micelles using an l-optimal design. The studies indicated that the MGF is released in 15 min by nano-micelles and has a particle size of 60 nm with a loading of more than 80%. The two breast cancer cell lines, i.e., MCF-7 and MDA-MB-231, were used in the cytotoxicity and cellular uptake experiments, which showed more kill and quicker cellular uptake from mangiferin-loaded mixed micelles. While in situ perfusion and in vivo pharmacokinetic studies suggested a nearly 3.0- and 6.6-fold increase in bioavailability and permeability of MGF after loading into the mixed micelles, the ex vivo intestinal permeability showed higher lymphatic uptake from the mangiferin-loaded mixed micelles as compared to drug alone. Hence, the pharmacokinetic performance of MGF in rats was improved by vitamin E-functionalized mixed micelles for increased anticancer potency [84].

Xiao et al. (2014) developed mangiferin-loaded magnetic microspheres using a modified solvent diffusion method. The study demonstrated improved cytotoxicity of MGF towards cancer cells after being loaded into magnetic microspheres, and the in vitro drug release experiments suggested that MGF would be successfully released from microspheres in a long way. As a result, the exhibited magnetite microspheres may have enormous potential as a successful MGF carrier [83].

Razura-Carmona et al. (2019) synthesized and optimized polymeric nanoparticles loaded with MGF using the emulsion solvent evaporation. The formulation’s polydispersibility index (PDI) value was 0.153, and the MGF encapsulation efficiency was 55%. MGF released gradually from 15 to 180 min showed high drug availability at the cancer site. According to the anti-topoisomerase assay results, the mangiferin-loaded polymeric nanoparticles displayed high anti-proliferative efficacy against prostate cancer cells compared to naïve MGF. The high doses of MGF had non-cytotoxic effects on BEAS 2B and HEPG2 cell lines. Finally, this study demonstrated an encapsulation process that exhibited in vitro gastric digestion resistance of 1.5 h without impairing healthy cells’ biological activity or metabolism [41].

Zhou et al. (2022) developed Tf-modified mangiferin-loaded solid lipid nanoparticles (Tf-MGF-SLNs) using the emulsification-solvent evaporation method. The Tf-MGF-SLNs had a mean hydrodynamic diameter of 121.8 ± 2.9 nm and a polydispersity index of 0.134. Tf-MGF-SLNs were discovered to have an entrapment efficiency of 72.4% and are spherical and uniform. The Tf-MGF-SLN drug accumulation release percentages reached 68% at pH 4.0 and 72% at pH 7.4 in 6 h, respectively, during the in vitro release exhibited controlled release of MGF from nanoparticles. The in vivo research revealed that Tf-MGF-SLNs, depending on the surface modification, suggested that cell internalization was altered and more MGF was successfully entered into the cancer cells. The Tf-MGF-SLNs effectively reduced tumor growth in the xenograft tumor model. The Tf-MGF-SLNs could be a promising formulation for treating lung cancer [101].

Alkholifi et al. (2023) prepared phospholipid-based mangiferin-loaded hydrogel for topical delivery in breast cancer. The drug entrapment was calculated to be more than 75% with a drug loading of 25% in the produced nanocarriers, whose globule size was smaller than 150 nm. After the Fickian drug release, the developed system offered a controlled release pattern. This increased MGF’s in vitro anticancer activity by a factor of four, and the MCF-7 cells’ cellular absorption was also improved by a factor of three. The ex vivo dermatokinetic studies demonstrated significant topical bioavailability with an extended residence duration. The research reveals a straightforward method for applying MGF topically, offering a safer, more potent and effective breast cancer treatment option. There may be a better alternative for today’s topical products of a traditional kind in the form of such scalable carriers with enormous topical delivery capability [82]. The mangiferin-based drug delivery systems based on the techniques used, the excipients of formulations and the outcomes have been described in Table 3 [102]. Therefore, all the research based on nano-drug delivery carriers of MGF on different cancers exhibited a promising hope for using the MGF in cancer after loading to various nanocarriers, which could enhance the drug’s therapeutic efficacy and safety profile.

### 4.3. Deep Learning-Based Approaches to Overcoming the Challenges

The difficulties posed by mangiferin in clinical translation can be significantly reduced by methods based on deep learning. Deep learning algorithms can analyze large datasets to create prediction models to help design and optimize mangiferin drug delivery systems based on nanotechnology. To forecast how well drug delivery systems will work, these models can consider several factors, including release kinetics, physicochemical features and nanoparticle characteristics [103]. These models can help researchers choose the most promising formulations and optimize medication release patterns by modeling various scenarios. Large databases of chemical compounds can be used to train deep-learning models to predict features like drug-likeness, solubility and bioavailability. By applying these models, researchers can find structural alterations or derivatization techniques that improve MGF’s stability, pharmacokinetic profile and therapeutic efficacy [104]. This can help develop brand-new mangiferin analogs or prodrugs with enhanced drug delivery properties. Toxicology data can be used to train deep learning models to anticipate probable adverse effects of mangiferin-loaded nanotechnology-based delivery devices. These models can offer insights into the safety profile of the formulations by incorporating pertinent physicochemical factors and nanoparticle features, assisting researchers in identifying potential toxicity issues early in the development process. Deep learning algorithms are excellent at analyzing multidimensional, complex data, including medical imaging [105]. They can be used to examine cancer patients’ imaging data to find particular tumor markers or traits that nanotechnology-based drug delivery systems can target. This may help create targeted treatments that improve mangiferin delivery to cancer cells while reducing unintended side effects. Deep learning algorithms can combine data types, including patient, clinical and genomic data, to find undiscovered links and patterns [106]. By mining these datasets, deep learning algorithms can discover unique insights into the mechanisms of action, biomarkers and patient stratification methods for medicines based on mangiferin. This can direct individualized treatment plans and raise the rate of clinical translation success. It is important to note that research is ongoing in using deep learning to overcome MGF’s limitations in clinical translation. Even if these methods appear promising, more research and validation are required to guarantee their dependability and usefulness in practical situations [107].

## 5. Clinical Trials and Patent Analysis of Mangiferin

MGF exhibits proven pharmacological effectiveness and drug-likeness properties as per Lipinski’s rule. Still, human clinical trials based on MGF or its derivatives as a therapeutic drug molecule are currently unavailable [108]. The patent survey also depicts that the scientists have used no standardized approach to evaluate the pharmacology of MGF. Some of the patents of MGF and its derivatives against cancer treatment have been mentioned in Table 4 [109].

## 6. Conclusions and Future Prospects

Several studies highlight the different biological effects of mangiferin and its mechanism of action toward many diseases. Hence, its low absorption and less bioavailability in the body are the main factors restricting its clinical use for cancer. In this regard, developing novel drug delivery systems for mangiferin has been a necessary field of research in recent times. The nanocarriers mentioned in this review have shown better therapeutic efficacy and quality of mangiferin against several types of cancers. Mangiferin was enclosed in nanoparticles as transporters, preventing its deterioration and facilitating its distribution to cancer cells. The controlled release features of these nanoparticles can be modified, allowing for sustained and precise medication administration. Nanocarriers such as micelles, dendrimers and nanoemulsions have also been investigated for the transport of mangiferin. These carriers may increase mangiferin’s solubility, stability and bioavailability, hence increasing its therapeutic potency.

Additionally, they can be functionalized with ligands or antibodies to target only cancer cells, enhancing medication accumulation and minimizing side effects. It is possible to add ligands or antibodies that precisely target cancer cell markers on the surfaces of nanoparticles and nanocarriers. By enhancing the targeted distribution of mangiferin to cancer cells, this surface modification raises the drug’s therapeutic index while lowering adverse effects on normal tissues. Despite the promises, there are several difficulties with mangiferin medication delivery using nanotechnology for cancer chemotherapy. Scaling up the production of nanocarriers and nanoparticles from the laboratory to industrial size can be quite challenging. Achieving the desired results takes a lot of effort and multiple optimizations. Clinical translation requires ensuring safety, reproducibility, scalability and quality control. The pharmacokinetics of mangiferin-loaded nanoparticles must be understood to optimize dosage schedules and assess potential toxicity. Extensive research is required to evaluate the safety and potential negative impacts of delivery systems based on nanotechnology. It is crucial to keep mangiferin and its delivery systems stable.

## Figures and Tables

**Figure 1 cancers-15-04194-f001:**
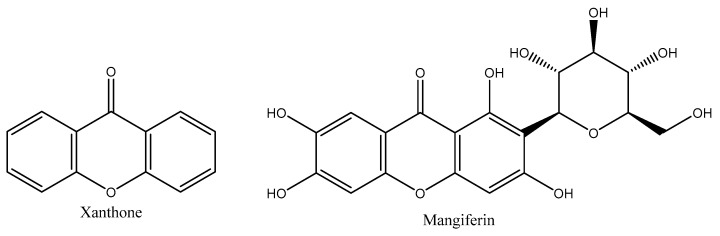
Chemical structure of xanthone and mangiferin.

**Figure 2 cancers-15-04194-f002:**
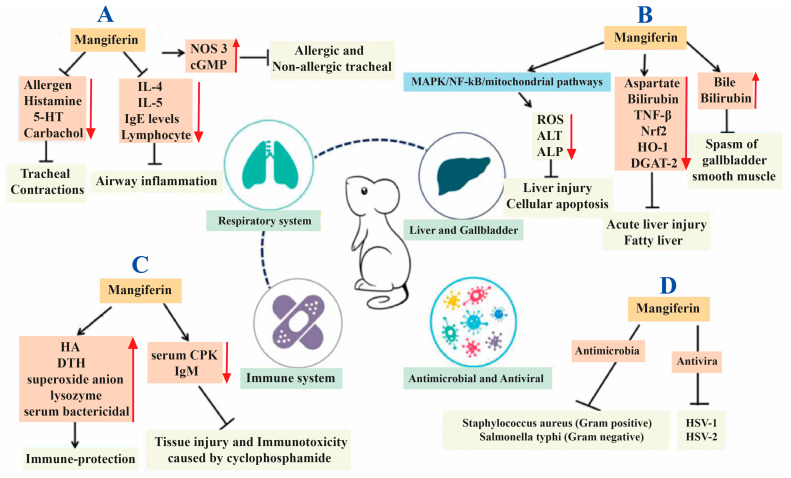
Representation of various actions of mangiferin on different disorders; (**A**) respiratory disorders, (**B**) gallbladder and liver diseases, (**C**) immunological disorders and (**D**) pathogenic microorganisms. IL: Interleukin; 5-HT: 5-hydroxytryptamine; NOS 3: Nitric oxide synthase 3; cGMP: Cyclic guanosine monophosphate; NF-κB: Nuclear factor-κB; MAPK: Mitogen-activated protein kinase; TNF-β: Tumor necrosis factor-β; Nrf2: Nuclear factor, erythroid like 2; HO-1: Heme oxygenase-1; DGAT-2: Diacylglycerol-acyltransferase 2; ROS: Reactive oxygen species; ALT: Alanine aminotransferase; ALP: Alkaline phosphatase; HA: Humoral antibody; DTH: Delayed-type hypersensitivity; CPK: Creatine phosphokinase; IgM: Immunoglobulin M; HSV: herpes simplex virus. Reproduced as per Creative Commons Attribution License from Du et al. [24].

**Figure 3 cancers-15-04194-f003:**
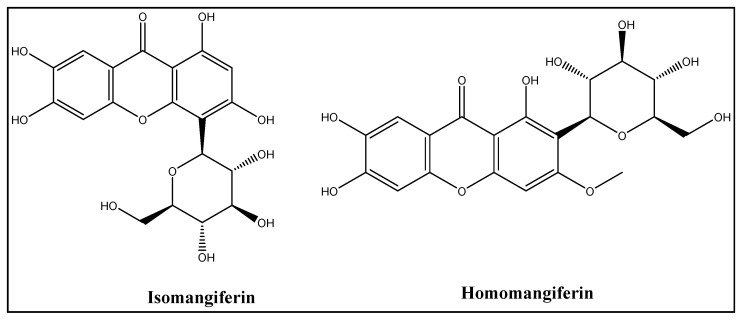
Chemical structures of isomangiferin and homomangiferin.

**Figure 4 cancers-15-04194-f004:**
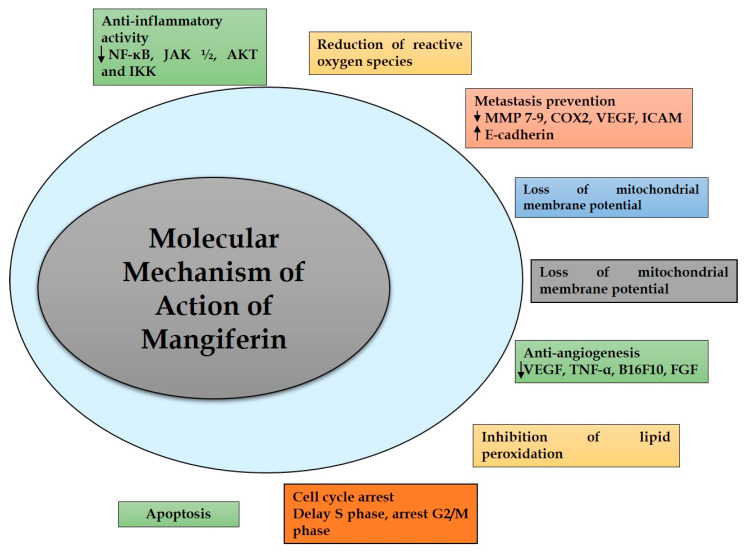
Molecular targets and biomarkers of mangiferin.

**Figure 5 cancers-15-04194-f005:**
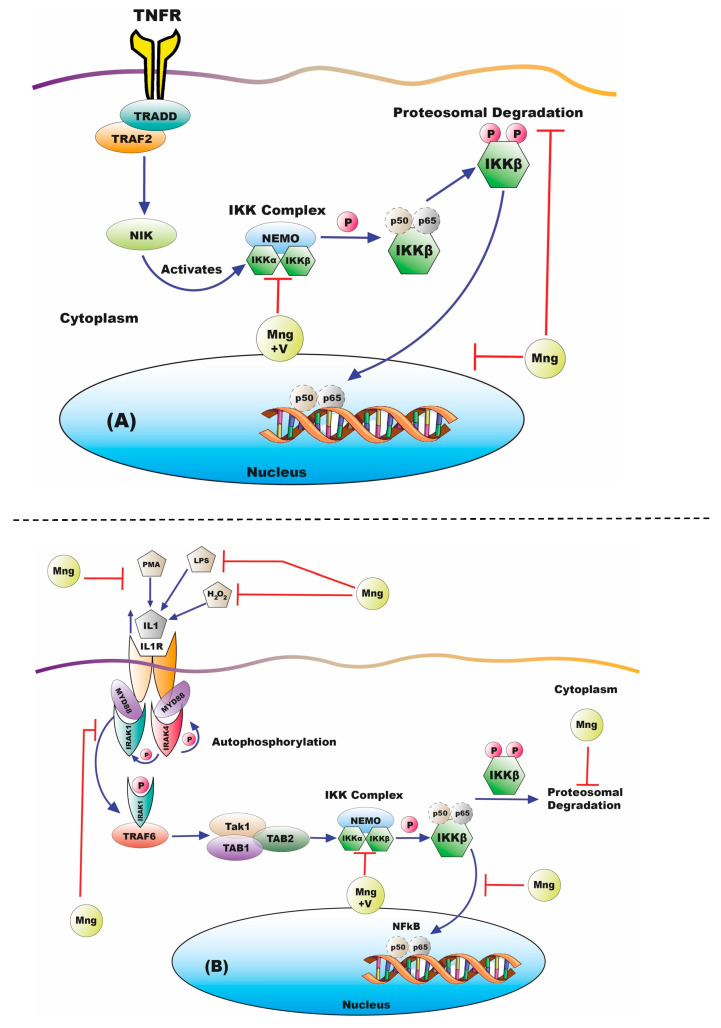
Inhibition of NFκB by mangiferin via (**A**) Classical (**B**) Alternative pathway. Mng: Mangiferin; V: Vimang^®^. Reproduced as per Creative Commons Attribution License from Gold-Smith et al. [55].

**Figure 6 cancers-15-04194-f006:**
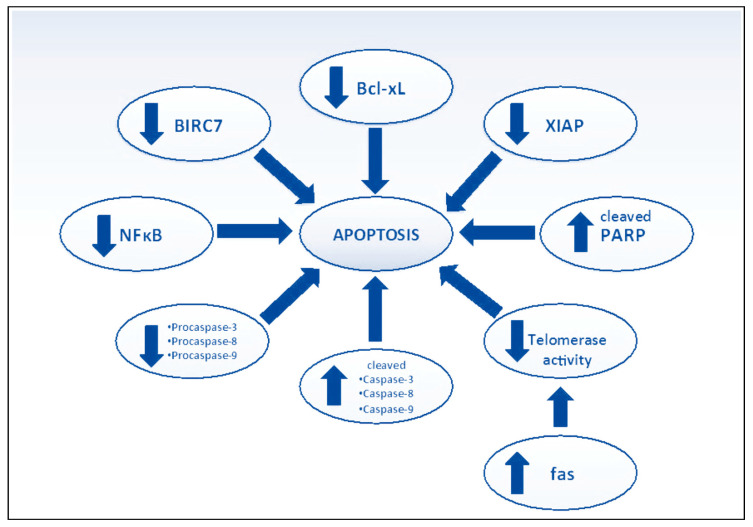
Effect of mangiferin on apoptosis. BIRC7: Baculoviral IAP repeat-containing protein 7; NF-κB: Nuclear factor- κB; Bcl-xL: B-cell lymphoma-extra-large; XIAP: X-linked inhibitor of apoptosis; PARP: Poly (ADP-ribose) polymerases; fas: Foreign Agricultural Service. Reproduced as per Creative Commons Attribution License from Gold-Smith et al. [55].

**Figure 7 cancers-15-04194-f007:**
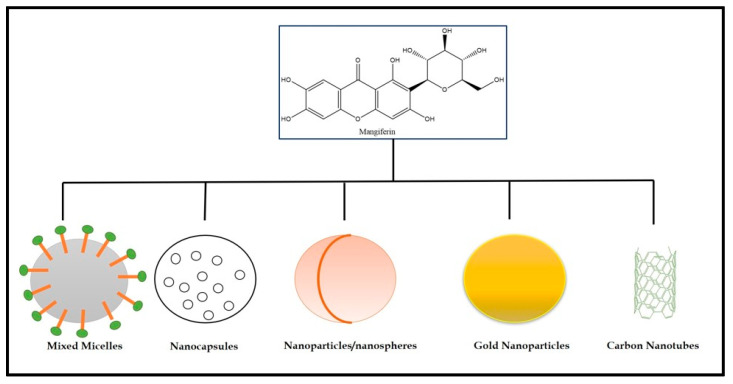
Types of nanocarriers used for the delivery of mangiferin to cancer cells.

**Table 1 cancers-15-04194-t001:** Therapeutic applications of different parts of the plant *M. indica*.

Plant Parts	Chemical Constituents	Therapeutic Applications
Fruits	Carotenes, xanthophyll esters, tocopherols and mangiferin	Prevent heat stroke, inflammation, prostate cancer, colon cancer, breast cancer and liver cancer
Stem bark	Mangifera indicasterol, manghopana, mangifera indica coumarin, mangifera indicaleanone and terpenoidal saponin indicoside A and B	Diabetes, anemia, menorrhagia, scabies, syphilis and cutaneous infections
Leaves	Catechin, alanine, mangiferin, tetracyclic triterpenoids, shikimic acid, protocatechuic acid and glycine	Scalds and dysentery

**Table 2 cancers-15-04194-t002:** Mechanism of mangiferin to cancer cells [38].

S. No.	Biochemical Markers	Targets	References
1.	Anti-inflammatory Potent inhibitor of CXCR4, ICAM1, NF-κB and XIAP	Nf-κB, AKT, IKK, STAT3, JAK1/2, IκB-α and MAPK	[66]
2.	ROS: Scavenge ROS present in cancer cells and inhibit xanthine oxidase	ROS	[73,74]
3.	Metastasis Inhibits activation of β-catenin and prevents the breast cancer	COX-2, VEGF, E-cadherin, ICAM and MMP7–9	[75]
4.	Antiangiogenesis Inhibit the growth of some tumors	TN-α, B16F10, FGF and VEGF	[68,76]
5.	Apoptosis Induces apoptosis by inhibiting NF-κB activation	Bcl-xL, Caspase 9, 7, 3, Bcl2 and XIAP	[77,78,79]
6.	Mitochondrial membrane potential	Induces loss of mitochondrial membrane potential and activates apoptotic proteins.	[80]

**Table 3 cancers-15-04194-t003:** The various types of mangiferin-loaded nanocarriers along with their excipients, method of preparation and outcomes [94].

S. No.	Formulations	Excipients	Techniques	Outcomes
1.	Mangiferin emulsion	Copolymer of ethylene vinyl acetate, vinyl acetate and toluene	Solvent evaporation technique	Increased antioxidant activity, increased tensile strength and increased mangiferin clearance.
2.	Mangiferin hydrogel	Polyvinyl alcohol, gelatin and chitosan	Sol-Gel technique	Controlled release of mangiferin from the matrix
3.	Mangiferin nanoemulsion	Hyaluronic acid, glycerin, water, lipoid S75 and trasncutol	Nanoemulsion technique	The average size of 296 nm, improved permeability and appropriate anti-inflammatory effect
4.	Mangiferin microparticles	Cellulose acetate phthalate	Supercritical antisolvent technique	Controlled release of mangiferin
5.	Mangiferin mixed micelles	Pluronic F127, vitamin E TPGS and pluronic P123	Thin-film hydration technique	Spherical morphology of micelles, high solubility and sustained release in the intestinal environment
6.	Mangiferin nanoparticles	Chitosan	Spray-drying technique	Accurate nano-size with cr(IV) removal pH-dependent release

**Table 4 cancers-15-04194-t004:** Patents of mangiferin and its derivatives.

S. No.	Compound Name	Application	Mechanism	Patent Number
1.	Norathyriol, tetraacetate	Prostate cancer	5α-reductase inhibitor	CN 104013611A
2.	Acetylated aglycone derivative I	Prostate cancer	5α-reductase inhibitor	CN 104013611A
3.	3-O-methyl-mangiferin acetate	Tumor	NA	CN 103755692A
4.	Mangiferin berberine salt	Anticancer	AMPK activator	WO 2010145192
5.	3-O-methyl-mangiferin benzoate	Tumor	NA	CN 103755693A

## Data Availability

The data used to contribute the findings of this review are included within the article.

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
