# Peer review of "Nanotechnology-Based Drug Delivery Approaches of Mangiferin: Promises, Reality and Challenges in Cancer Chemotherapy"

_cancers, 2023, doi:10.3390/cancers15164194_

Round 1

Reviewer 1 Report

This paper is a review that focuses on the potential of mangiferin (MGF) as a therapeutic drug for cancer. It provides a history and discovery of MGF, as well as its structure elucidation. The paper also discusses the mechanism of MGF in different types of cancers and recent approaches in MGF-based nanocarriers that have increased its solubility and bioavailability in cancer cells. The authors emphasize the need for further research to fully understand the potential of MGF as an effective anticancer agent. 

However, there are some major concerns:  The whole paper does not support enough evidence about the deep learning and its implications rather than some general ideas in a paragraph. Changing the title, therefore, is a must not to mislead the audiences. It also suffers from a relevant conclusion or a future direction. The quality of illustrations has to be improved, many of them are not readable due to font size or poor quality of figures. Figure legends do not explain the information of the figures properly. The cohesion and coherence have to be improved and the review needs to get re-structured as gathering some information and putting together is not a scientifc review.

Extensive editing of English language required. The cohesion and coherence have to be improved. The redundancy and repetitive statements affect readability.

Author Response

Comment 1: The whole paper does not support enough evidence about the deep learning and its implications rather than some general ideas in a paragraph. Changing the title, therefore, is a must not to mislead the audiences. It also suffers from a relevant conclusion or a future direction. The quality of illustrations has to be improved, many of them are not readable due to font size or poor quality of figures. Figure legends do not explain the information of the figures properly. The cohesion and coherence have to be improved and the review needs to get re-structured as gathering some information and putting together is not a scientific review. Extensive editing of English language required. The cohesion and coherence have to be improved. The redundancy and repetitive statements affect readability.

Response: The authors would like to express their gratitude to the esteemed reviewer for his insightful guidance. The title of the manuscript has been changed and incorporated in the revised manuscript. The conclusion or a future direction part has also been modified according to the directions of the respected reviewer. The figures' quality has been improved, and the figure's legends have been appropriately explained in the revised manuscript. Per the advice, the manuscript's cohesion, coherence and English language have also been improved. We have also checked the English language with Grammarly software. All the modifications are highlighted in yellow color.

Reviewer 2 Report

1) The quality of the English language should be substantially improved. For example: present review article (article is an extra word); with mangiferin in its applications in the nanosized versions (forms should be used instead of versions); Molecular Mechanism of Mangiferin in Cancer (is the word action missing?) etc. 2) The quality of the figures should be improved. I recommend that authors refer to the original source for a quality image, or adapt the figure (specify - adapted from Du et al. [24]) - leave the diseases discussed in this review. The same applies to figure 5. This figure is not readable. 3) The title of the article does not fully reflect the content of the article. Great attention has been paid to the anti-cancer mechanisms of pure mangiferin. Deep learning approach - only one section of the article is given. The title of the article should be reformulated.

The quality of the English language should be substantially improved. For example: present review article (article is an extra word); with mangiferin in its applications in the nanosized versions (forms should be used instead of versions); Molecular Mechanism of Mangiferin in Cancer (is the word action missing?) etc.

Author Response

Comment 1: The quality of the English language should be substantially improved. For example: present review article (article is an extra word); with mangiferin in its applications in the nanosized versions (forms should be used instead of versions); Molecular Mechanism of Mangiferin in Cancer (is the word action missing?) etc

Response: As per the advice, the quality of the English language of the manuscript has been improved using Grammarly software and incorporated in the revised manuscript. All the modifications are highlighted in yellow color.

Comment 2: The quality of the figures should be improved. I recommend that authors refer to the original source for a quality image, or adapt the figure (specify - adapted from Du et al. [24]) - leave the diseases discussed in this review. The same applies to figure 5. This figure is not readable.

Response: The figures' quality has been improved in the revised manuscript. All the modifications are highlighted in yellow color.

Comment 3: The title of the article does not fully reflect the content of the article. Great attention has been paid to the anti-cancer mechanisms of pure mangiferin. Deep learning approach - only one section of the article is given. The title of the article should be reformulated.

Response: The title of the manuscript has been changed and mentioned in the revised manuscript. All the modifications are highlighted in yellow color.

Reviewer 3 Report

This article consists of a total of 22 pages and is mainly divided into 6 parts. It focuses on the sources and structures of mangiferin (MGF) and its derivatives, as well as the benefits of this substance. It delves into the mechanism of action of MGF in different types of cancer, as well as recent research progress in successfully improving the solubility and bioavailability of MGF in cancer cells using nanocarriers based on MGF, as well as deep learning methods to overcome the challenge of effectively delivering MGF to the target site. This review also critically analyzes the future prospects of MGF as a potential candidate drug for cancer treatment. The topic of this paper is correct, the thinking is reasonable, and the theoretical argument is relatively rigorous. However, there were some details and English expression errors in the paper, and the summary comments are as follows:

Main viewpoints:

1.On the third page, it is mentioned in the text that when oral administration of MGF, the chemical composition of MGF meets the four standards of Lipinski'scriteria. Which four standards? Suggested supplementation

2.On page 9, the author mentions Vimang ® Extracts and MGF can inhibit TNF induced VEGF-A transcription in breast cancer cell lines. Is there any literature support? It is recommended to supplement

3.On page 9 and 3.5, the author mentioned that in vivo experiments, MGF also effectively reduced tumor volume in mice. Specifically, which type of tumor? Suggestion Description

4.On page 10, 3.7, Tan et al. found that MGF controls the inhibition of wt1 related lymphokine binding factor 1 (LEF1) through Wnt signaling, thereby inhibiting the growth of hepatocellular carcinoma. There are no references, it is recommended to add

5. On page 12, Figure 7 describes five types of nanocarriers used for delivering mangiferin to cancer cells. It is recommended to provide additional explanations on the differences and advantages of these five types of nanocarriers

Secondary viewpoint:

1.    Mangiferin appears for the first time in the abstract, and its abbreviation should follow

2. It is recommended to place the figure below the corresponding paragraph (as shown in Figure 1)

3. It is recommended that the annotations in Figure 2 briefly introduce the four small parts of ABCD (such as the small topic in one sentence)

4.On page 6, line 231 mentions 3A&B, but Figure 3 does not include A&B. It is recommended to modify it

5.The font and format of the references are not consistent. It is recommended to revise them according to the magazine's requirements

Author Response

Main viewpoints:

Comment 1: On the third page, it is mentioned in the text that when oral administration of MGF, the chemical composition of MGF meets the four standards of Lipinski's criteria. Which four standards? Suggested supplementation

Response: Per the suggestions, the four standards of Lipinski’s criteria have been mentioned in the revised manuscript. All the modifications are highlighted in yellow color.

Comment 2: On page 9, the author mentions Vimang ® Extracts and MGF can inhibit TNF induced VEGF-A transcription in breast cancer cell lines. Is there any literature support? It is recommended to supplement.

Response: The said reference has been added to the revised manuscript. All the modifications are highlighted in yellow color.

Comment 3: On page 9 and 3.5, the author mentioned that in vivo experiments, MGF also effectively reduced tumor volume in mice. Specifically, which type of tumor? Suggestion Description.

Response: The authors apologize for the typographical error. The corrections have been made in the revised manuscript. All the modifications are highlighted in yellow color.

Comment 4: On page 10, 3.7, Tan et al. found that MGF controls the inhibition of wt1 related lymphokine binding factor 1 (LEF1) through Wnt signaling, thereby inhibiting the growth of hepatocellular carcinoma. There are no references, it is recommended to add.

Response: The said reference has been mentioned in the revised manuscript. All the modifications are highlighted in yellow color.

Comment 5: On page 12, Figure 7 describes five types of nanocarriers used for delivering mangiferin to cancer cells. It is recommended to provide additional explanations on the differences and advantages of these five types of nanocarriers.

Responses: The five types of nanocarriers mentioned in Figure 7 have been described in the section “Need for Novel Drug Delivery Systems of Mangiferin” in the revised manuscript. All the modifications are highlighted in yellow color.

Secondary viewpoint:

Comment 1: Mangiferin appears for the first time in the abstract, and its abbreviation should follow.

Response: The said has been done in the revised manuscript. All the modifications are highlighted in yellow color.

Comment 2: It is recommended to place the figure below the corresponding paragraph (as shown in Figure 1).

Response: The said has been incorporated in the revised manuscript. All the modifications are highlighted in yellow color.

Comment 3: It is recommended that the annotations in Figure 2 briefly introduce the four small parts of ABCD (such as the small topic in one sentence).

Response: The legend of Figure 2 has been changed per the reviewer's advice. All the modifications are highlighted in yellow color.

Comment 4: On page 6, line 231 mentions 3A&B, but Figure 3 does not include A& B. It is recommended to modify it.

Response: The said modifications have been done in the revised manuscript. All the modifications are highlighted in yellow color.

Comment 5: The font and format of the references are not consistent. It is recommended to revise them according to the magazine's requirements.

Response: The references have been mentioned per the journal's guidelines in the revised manuscript. All the modifications are highlighted in yellow color.

Round 2

Reviewer 1 Report

No more comments.

Minor editing

Reviewer 2 Report

All my comments are taken into account. The article may be accepted for publication. Wish. . Conclusions. Abbreviations must be avoided.
